# An Exploration of the Influence of Non-Biomechanical Factors on Lifting-Related LBP

**DOI:** 10.3390/ijerph20031903

**Published:** 2023-01-20

**Authors:** Nic Saraceni, Amity Campbell, Peter Kent, Leo Ng, Leon Straker, Peter O’Sullivan

**Affiliations:** 1Curtin School of Allied Health, Curtin University, Bentley 6845, Western Australia, Australia; 2Department of Sports Science and Clinical Biomechanics, University of Southern Denmark Campusvej 55, 5230 Odense, Denmark; 3Body Logic Physiotherapy, Shenton Park 6008, Western Australia, Australia

**Keywords:** back pain, lift, manual handling, non-biomechanical

## Abstract

**Objective:** The primary objective was to compare non-biomechanical factors between manual workers with and without a history of LBP related to lifting. A secondary objective was to investigate associations between the change in pain intensity during repeated lifting (termed pain ramp) and non-biomechanical factors tested in the LBP group. **Methods:** Manual workers currently in lifting occupations with and without a history of lifting-related LBP were recruited (21 LBP and 20 noLBP) and took part in a repeated (100) lift task. A series of non-biomechanical factors, including psychological, work-related, lifestyle, whole health and psychophysical factors, were collected. Psychophysical factors (pressure pain thresholds (PPTs) and fatigue) were also measured at different time points. Associations between pain ramp during lifting and non-biomechanical factors were investigated with linear regression. **Results:** The LBP group reported worse perceived sleep quality, more musculoskeletal pain sites other than LBP and greater symptoms related to gastrointestinal complaints and pseudo-neurology compared to the group with no history of LBP. The group with LBP were also slightly more worried about the lifting task and felt more fatigued at the end of the lifting task. The feeling of fatigue during lifting was positively associated with pain ramp in the LBP group. Anxiety and gastrointestinal complaints were weakly negatively associated with pain ramp during lifting. **Conclusions:** The group differences of poorer perceived sleep, greater non-specific health complaints, slightly more worry about the lifting task and more perceived fatigue in the LBP group highlight the complex and multi-factorial nature of LBP related to lifting. The feeling of fatigue was positively associated with pain ramp in the LBP group, suggesting a close relationship with pain and fatigue during lifting that requires further exploration.

## 1. Introduction

The prevention of lifting-related LBP in manual occupations poses huge costs globally [1,2,3]. Workplace interventions have mainly had a biomechanical focus, with the aim to reduce lumbar spine forces/moments and flexion during loading [4]. Where lifting is unavoidable, engineering and administrative controls have been used to reduce lumbar force exposure, while workers in manual jobs are advised and trained to keep their back straight and squat to lift. Despite this significant investment, these interventions have largely proved unsuccessful in reducing the prevalence and burden of occupational LBP [2,5,6,7,8].

A recent systematic review compared the lifting biomechanics of people with and without LBP and reported that those with LBP followed current advice and lifted with less spinal range of movement, slower lifting velocities, greater knee bend and more trunk muscular activity than people without LBP [9]. However, the research assessed in that systematic review was of low methodological quality, limiting the certainty of the findings. A subsequent study addressed the limitations highlighted in this review and found that people with LBP demonstrated less intra-lumbar flexion, a deeper knee bend and slower lifting velocities compared to people with no LBP [10]. However, there were no convincing associations between the biomechanics of lifting and change in pain intensity during repeated lifting (termed “pain ramp” in this study), highlighting that other, non-biomechanical, factors may play a role in a person’s pain experience during lifting.

Non-biomechanical factors have been commonly reported to differ between groups with and without LBP. For example, studies that have prospectively investigated nurses and manual workers with and without LBP have reported that increased psychological distress [11,12], higher physical activity levels [12] and lower job satisfaction [13,14] are predictors of disabling LBP. Further, cross-sectional studies have identified factors such as negative back pain beliefs [15,16], poorer perceived sleep quality [17,18], more co-morbid health complaints [19] and greater sensitivity to psychophysical measures of pressure pain thresholds (PPTs) [20] in groups with LBP. These cross-sectional studies recruited participants from varied populations, such as nurses or nursing students, healthcare workers or the general population. It is unknown if these non-biomechanical factors are different in manual workers still engaged in lifting occupations, with and without a history of lifting-related LBP.

Prior biomechanical studies have failed to find convincing associations between lifting biomechanics and pain ramp, highlighting the potential role of non-biomechanical factors [10]. Studies that have investigated the direct influence of non-biomechanical factors on pain ramp during lifting are sparse. Sullivan et al. investigated relationships between depression, catastrophization and fear of movement on pain ramp during repeated lifting. They also investigated a surrogate of perceived fatigue (weight estimate of the object lifted). In that study, a chronic LBP population on disability payments recruited from a pain clinic performed two rounds of 18 lifts of canisters weighing 2.9–3.9 kg and reported pain intensity after every lift in one round and canister weight estimates in the other round of lifts (mean pain during lifting was 3.9/10). Those with higher levels of fear of movement experienced greater pain ramp up during lifting [21]. There was also a significant correlation between pain ramp and canister weight estimate. Those with greater pain ramp estimated higher object weights, suggesting a potential relationship between an indirect measure of fatigue and pain ramp in the LBP group. A study by La Touche et al. used the same lifting protocol in a chronic LBP cohort with moderate disability levels and found that lower pain self-efficacy (scores below 30 on the chronic pain self-efficacy scale) were related to greater pain ramp up during the lifting task [22]. Finally, Rabey et al. found greater pain ramp up during a task that incorporated 20 repeated pencil lifts from the floor to be weakly associated with greater sensitivity to cold and pressure stimuli and also greater emotional distress in a more disabled chronic LBP cohort [23]. In the above-mentioned studies, the lifting task was light, repeated less than 40 times and participants were not involved in manual work at the time, which limits the applicability of those studies to manual workers. It was also not reported if the included LBP groups had LBP that was related to lifting, as not all participants experienced an increase in LBP with the repeated lifting task and that may have affected the findings.

Two other studies explored pain ramp with repeated lifting and psychophysical measures [24,25]. In the study by Falla et al., groups with and without chronic LBP, recruited from the general population or local healthcare clinics, with low disability levels, completed 50 lifts of a 5 kg box, and PPTs were recorded at the lumbar spine both before and after the task. This study found that those with LBP were more sensitive to pressure than the group without LBP both before and after the lifting task, but did not explore fatigue. The other study by Kuithan et al. in a similar chronic LBP group, used a 60 lift task of a 5 kg box. The feeling of fatigue was recorded after each minute (Borg Scale 6–20). The group with LBP reported significantly more fatigue on average during the lifting task (control group 11.1/20 compared to the LBP group 13.1/20 *p* = 0.005). That study also found different PPT responses between groups, where the group with LBP did not experience the expected hypoalgesic response to pressure following the lifting task [25]. Together, these findings suggest that fatigue may play a role in the development of LBP with repeated lifting and augmented sensory processing and diminished inhibitory pain pathways may exist in LBP groups with lifting-related pain ramp. Neither of these two studies explored associations between PPTs or the feeling of fatigue and pain ramp during lifting.

To date, no study has investigated differences across a broad range of non-biomechanical factors, in manual workers with and without LBP, currently engaged in lifting occupations. Further, the lack of association between biomechanical factors and pain ramp during lifting highlights the need to investigate the role of non-biomechanical factors.

Therefore, this study aimed to explore the following:Differences in a range of non-biomechanical factors (psychological, work-related, lifestyle, whole health and psychophysical) between groups of manual workers with and without a history of lifting-related LBP;Associations between the change in pain intensity during lifting (termed pain ramp) and non-biomechanical factors tested in people with a history of lifting-related LBP.

## 2. Methods

This cross-sectional study investigated a range of psychological, work-related, lifestyle, whole health and psychophysical (tissue sensitivity) characteristics of manual workers with and without a history of LBP. Questionnaires were answered prior to a 100-lift task in a laboratory where LBP intensity was recorded after every 10 lifts. Pressure pain thresholds (PPTs) were measured before and after the lifting task. A “safety of round back lifting” single-item question was also asked following completion of the lifting task. These data were captured during a single 2-hour data collection session and the study received ethical approval from the Human Research Ethics Committee at Curtin University (HRE2018-0197). Biosymm (a third-party physiotherapy organization) supported this research by paying for participant vouchers worth AUD 50, but were not involved in any other component of this research.


**Sample**


Forty-two manual workers with and without a history of LBP were recruited by word of mouth, posters or direct recruitment at companies that have manual workers. All participants were >18 years of age, working in manual jobs that required regular lifting (>25 lifts/shift) for >20 h per week.


*LBP group selection criteria*


Must have reported axial LBP (between T12 and gluteal fold) for more than 3 months duration, where lifting was a primary aggravating factor (repeated lifting at work increased low back pain to a level of >3/10). Additionally, one of the following two criteria were also met: (i) at least 1 episode of LBP in the past 12 months where they were unable to attend work or they had to modify how or what they lift at work because of LBP or have taken medication for LBP or have seen a health practitioner for LBP; (ii) average weekly low back pain (past week) ≥ 3/10 [12,26]. Common exclusion criteria for biomechanical LBP studies were applied, such as radiculopathy, radicular pain, recent fracture/surgery, diagnosed systemic inflammatory disorder or other disorders that influenced lifting kinematics (such as severe hip and knee pain). People with a body mass index (BMI) over 28 kg/m^2^ were also excluded due to potential inaccuracies with data capture devices.


*noLBP Group Selection Criteria*


Must have had no history of LBP in the past 5 years and had been in manual work for that duration. Explicitly, this meant that the workers had not missed a day of work or altered activity levels due to LBP, and also had no LBP which exceeded 24 h in duration that was greater than 3/10 intensity on a numerical pain rating scale (NPRS) and had never visited a healthcare professional for LBP [27].


**Participant Characteristics**


Age (years), Body Mass Index (BMI) (kg/m^2^), biological sex and levels of pain on the NPRS (0–10), which included average pain over the week prior to data collection and pain pre-lifting task, were collected in both groups at the time of data collection.


*LBP Group Only*


LBP-related disability levels—The 24-item Roland Morris Disability Questionnaire (RMDQ) was used to capture disability levels in the LBP group, and is the most commonly used self-report instrument for measuring disability in this population [28]. The RMDQ scores range from 0 to 24, with higher scores indicating higher levels of disability [29].

Pain-related catastrophizing—The Pain Catastrophizing Scale (PCS) contains 13 items regarding past pain experiences and provides a total score, and three subscales assessing elements of catastrophizing which include rumination, magnification and helplessness [30]. For the purpose of this study, the PCS sum score was used, 0–52, with higher scores indicating greater pain catastrophizing. The PCS sum score has been shown to have excellent internal consistency, with coefficient alpha = 0.87. A score of above 30 is considered clinically relevant [31].

Pain-related fear of movement—Tampa Scale of Kinesiophobia (TSK) is a widely used measure of pain-related fear beliefs that contains 17 items that the participants rated on a four-point Likert scale, which ranges from “totally agree” to “totally disagree” [32]. Scores range from 17 to 68, where higher scores indicate greater fear. Scores of >36 are indicative of high pain-related fear [33,34].


**Psychological Factors**


The following were measured in all participants.


*Cognitive Factors*


Beliefs and attitudes about back pain—These were measured using the Back Pain Attitudes and Beliefs Questionnaire (BACK-PAQ). Scores range from 34 to 170, with higher scores indicating more unhelpful beliefs about the back. It has been shown to have acceptable internal consistency (α = 0.70; 95% CI 0.66 to 0.73), construct validity and test-retest reliability [35,36].

Self-efficacy—The Generalized Self-Efficacy Scale is a self-report measure of perception of the competence to cope with a broad range of stressful or challenging demands. It is scored from 10 to 40, with higher scores indicating greater self-efficacy [37]. The international average was reported to be 29.6 (based on over 20,000 scores from 25 countries), and it has item internal consistency between alpha 0.75 and 0.91 [38].

Beliefs about lifting with a round back—A single-item question to capture specific beliefs about lifting with a rounded back [39,40,41] was asked after completing the task: “How safe is it to lift with a bent back?”, where 0 was equivalent to “not safe at all” and 10 was considered as “extremely safe”.


*Emotional Factors*


Depression, Anxiety and Stress—The short-form of the Depression, Anxiety, Stress Scales (DASS-21) is a valid and reliable questionnaire with 3 subscales, each containing 7 statements evaluating depression, anxiety, and stress [42]. Each of the 21 statements are rated from 0 to 3, and the score is doubled to give a score of 0 to 42 points per subscale, with higher scores reflecting greater symptoms. A normal score is considered as 0–9 for depression, 0–7 for anxiety and 0–14 for stress. The DASS-21 has acceptable reliability and validity [43].

Lifting task-related worry—A novel single-item question was used in this study to evaluate lifting specific worry. This question was asked before the lifting task begun, “How worried are you about this lifting task?”, with 0 being “not worried at all” and 10 being “the most worried you’ve ever felt”.


**Work-Related, Lifestyle and Whole Health Factors**


Work satisfaction—A single-item workplace satisfaction question was asked, “On a scale from 0–10 how satisfied are you with your current job?”, where 0 was equal to “not at all satisfied” and 10 was “extremely satisfied”. This single-item question is both reliable and valid [44].

Physical activity levels—The self-report long form of the International Physical Activity Questionnaire (IPAQ) was used, which gives an estimate of physical activity levels over the past 7 days and is valid and reliable [45]. Physical activity levels across occupation, transport, household and leisure domains are included. Data across these domains were summed to indicate weekly averages (in minutes). Total sedentary time is also estimated within this questionnaire.

Sleep quality—The Pittsburgh Sleep Quality Index (PSQI) was used to assess self-rated sleep quality and disturbance over the previous month. It contains 17 questions examining sleep quality, quantity, disturbance and its effect on daily living. The scoring schema generates a final score from 0 to 21 points, with a score of >5 points suggesting significant sleep disturbance. This questionnaire been shown to be reliable and valid [46].

Whole health—The Subjective Health Complaints Inventory measures the occurrence of symptoms across 5 subscales: musculoskeletal, pseudo-neurology, gastrointestinal, allergy and flu-like [47]. It consists of 29 items and measures somatic and psychological complaints across these domains. Elements within each subscale include musculoskeletal pain (headache, neck pain, upper back pain, low back pain, arm pain, shoulder pain, migraine, leg pain), pseudoneurology (palpitation, heat flushes, sleep problems, tiredness, dizziness, anxiety, sadness/depression), gastrointestinal complaints (gas discomfort, stomach discomfort, diarrhea, obstipation, gastritis/ulcer, heartburn, stomach pain), allergy (allergies, breathing difficulties, eczema, asthma) and flu (cold/flu and coughing). For the purpose of this study, each subscale score was presented separately. We removed LBP from the musculoskeletal component in order to reflect musculoskeletal pain across other body regions.


**Psychophysical Factors**


Tissue sensitivity—Pressure Pain Thresholds (PPTs) were defined as the point at which the sensation of pressure perceived by the participant changed from pressure alone to a sensation of pressure and pain [48]. PPTs were tested using an algometer with a probe size of 1 cm^2^ (Somedic AB, Hörby, Sweden). Four threshold measurements were taken at a ramp rate of 50kPa/s, with the mean of the last three thresholds used for analysis. Participants were measured before and after undertaking the lifting protocol at both the low back (over the point of most pain in the LBP group and 5 cm lateral to the L4/5 inter-spinous space on the right side over the erector spinae in the noLBP group) and also the dorsum of the right wrist in both groups. The left wrist was only used if there was a history of pain or trauma to the right wrist [49].


**The Lifting Task, Fatigue and Pain Ramp Measures**


The lifting task, which comprised 100 lifts, has been described previously [10] and was adapted from a previously published protocol [26]. Levels of pain intensity and fatigue were captured before and during the lifting task without the participant pausing the task. An explanation was given prior to the lifting task that LBP on a 0–10 NPRS would be asked every 10 lifts by a researcher during the repeated lifting task. A 0 score was reflective of no LBP and 10 was the worst LBP imaginable. After every 10 lifts, the researcher asked, “What is your level of pain 0–10?” The feeling of fatigue was also asked (0–10 modified Borg Scale [50]), where 0 was not fatigued and 10 was indicative of maximal fatigue.


**Data analysis**


The demographic data were analyzed using Chi Square analysis for sex and bias-corrected bootstrapped (100 samples with replacement) linear regression for other demographic, pain and fatigue variables. Where descriptive data were not normally distributed, medians and inter-quartile ranges were reported.

For the variables of primary research interest, all but one of the between-group comparisons were analyzed with bias-corrected bootstrapped (100 samples with replacement) linear regression, with group (LBP/NoLBP) being an independent variable. Both unadjusted and adjusted (for age and sex) estimates were reported for all those analyses. The exception was the between-group comparison of IPAQ physical activity categories (high/moderate/low), which was analyzed using Fisher’s Exact Test.

To identify whether pain ramp (10 measurements) during the lifting task was associated with questionnaire data (1 pre-lift measurement) or PPT data (1 change score), pain ramp slope was the dependent variable and a questionnaire or PPT score was the independent variable. Pain ramp slope (degrees) was the average of each individual’s change in pain intensity over the 100 lifts, as calculated by that individual’s regression slope in a preliminary linear model where pain intensity was regressed by lift decile. To identify any association between pain ramp and fatigue during the lifting task, the 10 measurements of pain (measured at each decile of lifts) were the dependent variable and the 10 corresponding measurements of fatigue were independent variables.

Questionnaire data were captured using Qualtrics and exported via Excel to STATA version 15.1 (StataCorp, College Station, TX, USA) in which all analyses were performed. As this was an exploratory study, a *p* value of < 0.05 was used as the threshold for statistical significance, with no adjustment for multiple testing.

There was a small amount of missingness in these data. The data from one female in the noLBP group were removed from the analysis within a related biomechanical study because her BMI was an outlier, and consequently, her non-biomechanical data were also removed from the current analysis. In the non-biomechanical data, there was only the following missingness: the BACK-PAQ (2 of 41 participants), “How safe is flexed back lifting?” single item (3/41), DASS-Depression (1/41), DASS combined (1/41) and Sleep Quality (1/41), and those between-group comparisons were conducted only on participants who had no missing data. Similarly, in the non-biomechanical associations with pain ramp, only 2 of 22 analyses had any missing data (1 LBP participant missing in the BACK-PAQ and 3 LBP participants missing in “How safe is flexed back lifting?” single item. Due to this low level of missingness, no imputation was performed.


**Sample Size**


Data from 41 participants (LBP = 21, NoLBP = 20) were included in the between-group regression analyses. The *STATA* power oneslope command indicated that with 41 observations, there was a power of 80% to detect an effect size of 0.45 with a two-sided test at a 5% significance level, assuming a standard deviation of 1.0 for the covariate and the error.

## 3. Results


**Participant Characteristics**


The data of 21 LBP and 20 noLBP participants were analyzed. There were no differences between groups for age, sex and BMI. For the measures that were LBP group-specific, both pain catastrophizing and low back-related disability were low, whereas fear of movement was just below the threshold for high fear (Table 1).

Measures of LBP intensity before, during and at completion of the lifting task were significantly different between groups (Table 2).


**Between Group Comparisons of Non-Biomechanical Measures**


All of the following refer to the results in Table 3.


*Cognitive Factors*


There were no group differences based on the cognitive measures. Both groups had negative beliefs and attitudes about the back and reported that round back lifting was not safe. Self-efficacy was high in both groups.


*Emotional factors*


The only difference between groups from all the psychological variables was the pre-lift worry single item, although both groups reported very low levels of worry (LBP 0.9/10 vs. noLBP 0.1/10) about the lifting task. Levels of depression, anxiety and stress were within normal ranges in both groups.


*Work-Related, Lifestyle and Whole health Factors*


The PSQI global score for perceived sleep quality was higher in the LBP group, signifying perceived worse sleep. The LBP group also scored higher on musculoskeletal pain in regions other than low back, as well as gastrointestinal and pseudoneurology complaints sub-sections of the Subjective Health Complaints Inventory. Both groups scored equally high on IPAQ measures of physical activity, related to having highly physical occupations, and both groups were satisfied with work.


*Psychophysical*


Pressure pain thresholds (PPTs) measured prior to and after the lifting task, as well as change in PPTs (comparing pre- vs. post-lifting task PPTs), were not different between groups at any stage at either the lumbar spine or wrist. Both groups demonstrated similar hypoalgesic responses to pressure following the lifting task (i.e., less sensitive to pressure at the wrist and lumbar spine after the task). The PPTs for both groups were comparable to known reference thresholds in young people without pain at both the lumbar spine and wrist (Waller et al., 2016).


**Associations Between Pain Ramp and Non-Biomechanical Measures in the LBP Group**


Of the 23 non-biomechanical variables, three showed an association with pain ramp after adjustment for sex and age. The DASS—Anxiety subscale and the Subjective Health Complaints Inventory—Gastrointestinal subscale were both negatively associated with the positive slope of the pain ramp (Table 4).

When exploring the relationship between pain on lifting (pain ramp) and fatigue during lifting, there were three statistically significant effects (Table 5). The effect of fatigue during lifting was that for every 1.0 point increase in fatigue, pain increased by 0.240 points more in the pain group than if the same increase in fatigue was experienced by the noLBP group. Moreover, on average, the effect of being in the LBP group was 1.298 points more pain during lifting. Lastly, pain during lifting increased by 0.010 points with every 10 lifts over time.

## 4. Discussion

This study explored differences between manual workers with and without a history of LBP, across a range of non-biomechanical variables previously reported to be either risk factors for or associated with LBP. People in the LBP group reported poorer perceived sleep quality, greater symptoms in three elements of the Subjective Health Complaints Inventory (musculoskeletal pain other than LBP, pseudoneurology and gastrointestinal complaints) and were slightly more worried about the lifting task. The group with LBP also felt more fatigued at the end of the lifting task.

We also explored associations between non-biomechanical variables and pain ramp during the lifting task in the LBP group. The feeling of fatigue during lifting was positively associated with pain ramp in the LBP group. Anxiety and gastrointestinal complaints were weakly negatively associated with pain ramp during lifting (i.e., higher anxiety scores and more gastrointestinal complaints were associated with less pain ramp up during lifting).


**
*Aim 1—Non-Biomechanical Differences between Groups*
**


The finding of poorer perceived sleep in the LBP group is consistent with a growing body of research which supports that poor perceived sleep quality predicts the onset, persistence and worsening of LBP, as well as sickness absence, in varied LBP cohorts [17,51,52,53,54]. For example, in a prospective study of hospital workers (which included nurses, physiotherapists and doctors), poorer perceived sleep was reported to increase the risk of the development and worsening of LBP [17]. Further, in a large cross-sectional study of nurses [18], perceived poorer sleep significantly increased the odds of disabling LBP. However, it is not known if lifting was specifically aggravating for the LBP group in those studies. Given the strength and consistency of the evidence in longitudinal studies implicating perceived poor sleep to be a risk factor for LBP, further investigation of how sleep may influence LBP in manual workers is warranted. Of note, there remains a point of debate as to how closely perceived sleep quality reflects actual sleep (sleep efficiency, latency, total sleep time, etc.), and therefore, future longitudinal studies investigating the relationship between sleep and pain should consider capturing both the perception of sleep and objectively measured sleep (e.g., by polysomnography or actigraphy) [55,56].

People in the LBP group reported more co-occurring musculoskeletal pain, including headache, neck pain, upper back pain, arm pain, shoulder pain, migraine and leg pain. This finding is consistent with previous research in large general population survey studies that found localized LBP to be rare [57,58]. This study adds to the known relationship between LBP and the co-occurrence of multiple co-morbidities, including other musculoskeletal and non-musculoskeletal pain [59]. Importantly, a higher number of pain sites show a linear association with more severe physical, psychological and social impact in previous literature [60,61,62]. The report of greater pseudoneurology and gastrointestinal complaints in the LBP group in our study is a novel finding in lifting-related LBP but is in-line with previous reports of a close relationship between greater musculoskeletal and non-musculoskeletal complaints, implicating the potential for common mechanisms driving more widespread sensitization [52,63,64,65,66].

This was the first study to investigate if perceived fatigue was different during a repeated lifting task in manual workers with and without a history of LBP. Previous studies investigating fatigue during lifting have mainly compared stoop and squat-style lifting in people without LBP [67,68,69,70]. In one study that compared fatigue at the end of a repeated lifting task, the group with chronic LBP also reported significantly higher fatigue [25]. In the study by Kuithan et al. the chronic LBP group members were not manual workers and whether lifting was an aggravating factor was not known (fatigue on average control group 11.1/20 compared to the LBP group 13.1/20 *p* = 0.005). Feeling fatigued is commonly associated with persistent LBP and is potentially influenced by a range of physical (strength and endurance, levels of conditioning) and non-physical factors (pain, perceived sleep quality, mood) [71,72]. Future lifting studies in people with LBP should explore both physical vs. non-physical drivers of increased perceived fatigue.

Due to the cross-sectional design of this research, the temporal/causal directions of the relationships between perceived poor sleep, non-specific health complaints, musculoskeletal pain and fatigue are not known. However, previous research suggests a complex inter-relationship between some of these factors [73]. For example, poor sleep is thought to increase circulating levels of pro-inflammatory cytokines and cortisol [74] and also alter dopamine and opioid-related endogenous pain modulatory systems [73,75]. This occurs in combination with other complex alterations to the hypothalamo–pituitary–adrenal axis and other spinal and supraspinal sensory processing changes in those with persistent pain [76,77,78]. Together, these alterations of the neuro-endocrine-immune system may be the common mechanism linking poor sleep, multiple musculoskeletal pain sites, non-specific health complaints and fatigue that often co-occur and have bi-directional relationships [47,54,59,64,67,79].

Interestingly, there were no differences in PPTs between the LBP and noLBP groups, both locally at the lumbar spine and also remotely at the wrist prior to or after the lifting task. Further, both groups in our study demonstrated hypoalgesic responses (less sensitive to pressure) after the lifting task, suggestive of normal descending pain inhibitory pathways [80]. Two other studies have compared PPTs before and after a lifting task in groups with and without LBP. The findings of both of these studies were different from our results. The first study by Falla et al. reported that a chronic LBP group was more sensitive than the noLBP group in the lumbar spine before and after the lifting task. Another study by Kuithan et al. reported that the group with chronic LBP had similar lumbar PPTs to the noLBP group before the lifting task, but only the group without LBP experienced hypoalgesic responses to pressure following the task. There are many possible reasons why the hypoalgesic response to pressure following the lifting task was different in the LBP group in our study. In the Kuithan et al. study, the LBP group had a higher pain intensity (3.9/10 prior to starting lifting task), and in both of the other studies, pain catastrophizing (mean PCS 14.5 and 14.9/52) was higher compared to the LBP group in our study (median score of 8 (5–14)). Higher baseline pain intensity and greater pain catastrophizing have been demonstrated to alter PPTs and hypoalgesic responses to exercise [81,82]. The PPT protocols across studies were also different, which may have contributed to the disparate findings. In the Kuithan et al. study, the mean of 16 locations across the lower back was reported, and eight sites were reported in the study by Falla and colleagues, whereas we used the site of most LBP as indicated by the participant. Baseline PPTs were much higher in our study, suggestive of a lower sensitivity profile to pressure in our LBP cohort. Finally, our lifting protocol was of greater intensity (more repetitions and heavier box weight). Exercise of higher intensity has been shown to induce greater hypoalgesic responses on average, in both LBP and healthy cohorts in a meta-analytical review of responses to exercise [80]. However, as the sample sizes in all three studies were small, and the known variance of PPT data is large, the differences between studies may be a chance finding. Future studies should have bigger samples and also a greater battery of quantitative sensory testing to better understand the sensitivity profiles of groups with and without LBP related to lifting.

The results regarding the cognitive factors revealed no meaningful differences between groups. Both groups had negative back pain beliefs and thought that round back lifting was less safe than straight back lifting, while the LBP group members were slightly more worried about the lifting task. These findings are consistent with previous research and highlight the negative beliefs that most people hold towards LBP [15,16,35] and round back lifting [39,40,41,83,84,85]. This may be a reflection of occupational lifting training and advice received, or information from previous encounters with health care practitioners [39,40,41,86]. There were also no differences between groups in their self-rated emotions (depression, anxiety and stress), which were all in the low range, suggesting that both groups had good mental health. This finding of a normal self-reported emotional state is consistent with LBP groups with lower levels of pain and disability and higher self-efficacy [22].


**
*Aim 2—The Association between Pain Ramp and Non-Biomechanical Factors*
**


Pain ramp up during a repeated lifting task has been reported to be of greater influence on disability levels than static/once off pain intensity measures [87]. In our study, the LBP group averaged a 2.2 point increase in pain (3.8/10 at the end of the task) on the NPRS, which is considered clinically meaningful [88]. Other studies investigating pain ramp up during a lifting task have reported less pain ramp on average, but used lighter objects and fewer repetitions [21,24,25]. Of the 23 non-biomechanical measures explored for associations with pain ramp, the feeling of fatigue was the most convincing and showed a positive association with pain ramp in the LBP group. This result is supported in a previous study by Sullivan et al. where a surrogate of perceived fatigue (greater canister weight estimate) and pain ramp in a more disabled LBP group were correlated (r = 0.23 *p* < 0.05). Given that people with LBP commonly report heightened levels of fatigue as a factor in the onset of LBP [89] and in LBP persistence [90], future lifting studies should explore fatigue and lifting-related LBP in a more detailed manner. For example, it is not known from this study if the report of greater fatigue reflects muscular deconditioning or a whole-person perception of fatigue (e.g., related to poor sleep or pain persistence). Clarifying the relationship between the elements of fatigue that influence pain ramp during lifting may provide insight into future intervention targets.

Anxiety and gastrointestinal complaints were weakly negatively associated with pain ramp. However, the clinical relevance of those findings is limited by the low levels of anxiety and gastrointestinal complaints in the LBP group. These findings are also at odds with previous research. For example, Sullivan et al. found higher levels of fear of movement to be correlated with greater pain ramp up during repeated lifting in a work-disabled LBP group receiving disability benefits [21]. However, mean fear of movement was much higher in that study (43 vs. 36 on the TSK). In another study, greater pain ramp up was related to a chronic LBP sub-group with a combination of heightened psychological distress and also greater sensitivity to pressure and temperature stimuli [23]. Our study found contradictory findings which may be related to the LBP cohort being less disabled with lower pain intensity levels and that we investigated singular non-biomechanical associations with pain ramp, and did not combine non-biomechanical factors in a multivariate analysis. Previous authors have concluded that pain ramp up during lifting is likely complex and highly individualized [23,91].


**Limitations**


This study was cross-sectional; prospective longitudinal research is needed to understand the directionality of the findings. Most of the non-biomechanical measures were by questionnaire, where a self-preservation bias may exist. When measuring factors such as perception of sleep quality or psychological stress, these perceptions may not be true reflections of actual sleep quality or stress biomarkers. This study should be considered exploratory and thus incorporated a large number of comparisons which might have inflated the risk of type 1 error.

An important distinction of this study is the LBP group that was recruited. The participants were employed in manual occupations (>20 h per week) that involved repeated lifting and at the time of recruitment were not care-seeking. They were recruited on the basis of greater than ≥3/10 pain related to lifting and at least one episode of LBP in the past 12 months where they were unable to work or where they sought care for their LBP. The LBP group profile was one of relatively low levels of pain, disability, negative back pain beliefs, moderate levels of pain-related fear, high self-efficacy and good mental health. They were also highly physically active and had healthy BMI scores. Therefore, the findings of this study may not be generalizable to people with LBP who are not of a similar profile. For example, those who are care-seeking, not working and who have higher levels of pain, disability and pain-related distress.

It is unknown if these results can be generalized to age groups that are different from those in our sample.


**Implications**


While this study is cross-sectional, and therefore, temporality is not known, some of the non-biomechanical factors identified in this study (such as perceived poor sleep quality/quantity, including sleep disturbance) have consistently been prospectively associated with the development of LBP and LBP persistence. The finding of poor sleep combined with the presence of musculoskeletal and non-specific health complaints and greater fatigue levels lends support to previous research suggesting that persistent LBP is related to complex neuro–endocrine–immune system alterations.

Intervention studies that have targeted multiple factors associated with a person’s LBP (such as negative back pain beliefs, fear avoidance behaviors and lifestyle factors such as sleep) have shown some encouraging outcomes, with reductions in LBP-related disability and also concomitant improvement in non-biomechanical factors (e.g., pain-related fear, pain self-efficacy and sleep quality) [92,93,94]. Specifically targeting non-biomechanical factors, such as sleep quality, in a cohort with lifting-related LBP has not yet been trialed. However, in a large prospective cohort study of 1777 people, perceived sleep improvements have been shown to be predictive of the resolution of multi-site pain at 5-year follow-up, suggesting its potential importance in both occupational LBP prevention and management strategies [52].

The association between the feeling of fatigue and pain ramp during lifting in people with lifting-related LBP highlights the need for more research to understand the drivers of LBP with repeated loading of the spine. Future studies could consider exploring contributors of greater perceived fatigue and also use a larger battery of quantitative sensory testing measures, such as the inclusion of cold pain thresholds, temporal summation and cold pressor testing. Together, this testing may provide further insight into those who may experience greater pain during repeated lifting.

## 5. Conclusions

This exploratory study compared a broad range of non-biomechanical factors between manual workers with and without a history of LBP. The LBP group members were currently working in a highly physical occupation and reported worse sleep quality, more musculoskeletal pain sites other than LBP and greater symptoms related to gastrointestinal complaints and pseudoneurology compared to the group with no history of LBP. The group with LBP were also slightly more worried about the lifting task and reported greater fatigue by the end of the lifting task. There was a clear association between fatigue and pain ramp up in the LBP group. No other associations between singular non-biomechanical variables and pain ramp during the 100-lift task were convincing. These findings highlight the complex and multi-factorial nature of LBP related to lifting and the need for further high-quality longitudinal studies.

## Figures and Tables

**Table 1 ijerph-20-01903-t001:** Participant characteristics of workers with (LBP) and without (noLBP) a history of lifting-related low back pain.

	LBP (n = 21)	noLBP (n = 20)	*p* Value ^#^
**Sex, Female n (%)**	7/21 (33.3%)	6/20 (30%)	0.819
**Age (years) mean (95%CI)**	37.5 (31.0–44.1)	32.3 (27.4–37.3)	0.169
BMI (kg/m^2^) mean (95%CI) Low back related disability (RMDQ 0–24 Scale) median (IQR)	24.0 (23.0–25.1) 5 (4–7)	24.0 (22.7–25.4)	0.993
Fear of movement (TAMPA 17–68 Scale) median (IQR) Pain catastrophizing (PCS 0–52 Scale) median (IQR)	36 (31–42) 8 (5–14)		

^#^ Participant characteristics were compared with Chi Square analysis for sex and bias-corrected bootstrapped (100 samples with replacement) linear regression for age and BMI.

**Table 2 ijerph-20-01903-t002:** Pain intensity and fatigue of workers with (LBP) and without (noLBP) a history of lifting-related low back pain.

	LBP (n = 21)	noLBP (n = 20)	*p* Value ^#^
**Pain—Average previous week (0** **–** **10 Scale)**	3.5 (2.7–4.3)	0.3 (0.0–0.6)	**<0.001**
**Pain—Entering lab (0** **–** **10 Scale)**	1.9 (1.3–2.6)	0.4 (0.1–0.7)	**<0.001**
**Pain—Beginning of lifting task (0** **–** **10 Scale)**	1.6 (1.0–2.3)	0.1 (−0.0–0.3)	**<0.001**
**Pain—End of lifting task (0** **–** **10 Scale)**	3.8 (2.7–4.9)	0.8 (0.3–1.3)	**<0.001**
**Fatigue—Post-lifting task (0** **–** **10 Scale)**	6.5 (5.5–7.4)	3.3 (2.0–4.6)	**<0.001**

# Between-group comparisons of pain and fatigue variables were analyzed with bias-corrected bootstrapped (100 samples with replacement) linear regression.

**Table 3 ijerph-20-01903-t003:** Comparison of non-biomechanical measures for workers with (LBP) and without (noLBP) a history of lifting-related low back pain.

	Group Values (95%CI) (Unadjusted)	Difference (Unadjusted)	Difference (Adjusted *)
** *Cognitive* **			
**Back Pain Attitudes and Beliefs (BACK-PAQ 34–170 scale)**	LBP 111.2 (105.5 to 116.9) noLBP 110.5 (105.5 to 115.6)	0.7 (−7.1 to 8.4) *p* = 0.865	1.2 (−6.4 to 8.9) *p* = 0.753
**How safe is flexed back lifting single item (0–10 scale (0 = not safe at all))**	LBP 2.7 (1.6 to 3.9) noLBP 3.1 (2.0 to 4.2)	−0.4 (−2.0 to 1.2) *p* = 0.644	−0.6 (−2.3 to 1.0) *p* = 0.452
**Self-Efficacy (GSEQ 10–40 scale)**	LBP 34.7 (33.2 to 36.2) noLBP 32.5 (30.8 to 34.1)	2.3 (−0.0 to 4.5) *p* = 0.050	2.2 (−0.2 to 4.6) *p* = 0.078
** *Emotional* **			
**Subjective worry pre—lift task (0–10 Scale)**	**LBP 0.9 (0.2 to 1.7)** **noLBP 0.1 (−0.1 to 0.4)**	**0.8 (0.0 to 1.6)** ***p* = 0.038**	**0.9 (0.1 to 1.7)** ***p* = 0.033**
**Depression (DASS 21 0–42 scale)**	LBP 7.1 (5.0 to 9.3) noLBP 5.0 (2.7 to 7.4)	2.1 (−0.9 to 5.1) *p* = 0.178	2.1 (−1.2 to 5.3) *p* = 0.212
**Anxiety (DASS 21 0–42 scale)**	LBP 6.8 (4.8 to 8.7) noLBP 4.7 (2.2 to 7.2)	2.1 (−1.2 to 5.3) *p* = 0.212	2.2 (−1.7 to 6.1) *p* = 0.269
**Stress (DASS 21 0–42 scale)**	LBP 10.8 (4.8 to 8.7) noLBP 8.2 (2.2 to 7.2)	2.6 (−1.3 to 6.4) *p* = 0.191	2.3 (−1.5 to 6.2) *p* = 0.237
**DASS combined score (0—126 scale)**	LBP 24.7 (19.1 to 30.2) noLBP 18.5 (11.4 to 25.7)	6.1 (−2.5 to 14.8) *p* = 0.164	5.6 (−3.7 to 15.0) *p* = 0.237
** *Work-related, lifestyle and whole heath* **			
**Sleep Quality (PSQI 0–21)**	**LBP 6.0 (5.0 to 7.1)** **noLBP 4.6 (3.8 to 5.5)**	**1.4 (0.1 to 2.7)** ***p* = 0.034**	**1.7 (0.2 to 3.1)** ***p* = 0.021**
**Health Complaints –Modified Musculoskeletal (LBP excluded) (0–21)**	**LBP 4.4 (2.9 to 5.9)** **noLBP 1.2 (0.7 to 1.8)**	**3.2 (1.6 to 4.7)** ***p* < 0.001**	**3.4 (1.7 to 5.1)** ***p* < 0.001**
**Health Complaints—Pseudoneurology (0–21)**	**LBP 3.7 (2.6 to 4.7)** **noLBP 1.8 (0.9 to 2.6)**	**1.9 (0.5 to 3.2)** ***p* = 0.007**	**2.2 (1.0 to 3.5)** ***p* < 0.001**
**Health Complaints—Gastrointestinal (0–21)**	LBP 2.4 (0.9 to 3.9) noLBP 0.8 (0.2 to 1.5)	1.5 (−0.2 to 3.2) *p* = 0.076	**2.0 (0.3 to 3.7)** ***p* = 0.019**
**Health Complaints—Allergy (0–15)**	LBP 2.4 (0.9 to 3.9) noLBP 0.0 (−0.4 to 0.1)	0.2 (−0.1 to 0.6) *p* = 0.170	0.3 (−0.1 to 0.7) *p* = 0.199
**Health Complaints—Flu (0–6)**	LBP 0.7 (0.1 to 1.2) noLBP 0.9 (0.2 to 1.6)	−0.2 (−1.1 to 0.6) *p* = 0.600	−0.3 (−1.2 to 0.6) *p* = 0.468
**Physical Activity Levels (IPAQ—Total METmins/week)**	LBP 16233 (13315 to 19151) noLBP 20827 (13041 to 28612)	−4593 (−12979 to 3792) *p* = 0.283	−5689 (−14273 to 2894) *p* = 0.194
**Physical Activity Levels—IPAQ Categorical score High/Moderate/Low Scale)**	LBP→noLBP 20/21 High→19/20 High 1/21 Low→1/20 Mod	*p* = 1.000	
**Sedentary time/week—IPAQ (mins)**	LBP 1872 (1459 to 2286) noLBP 1874 (1481 to 2268)	−2.6 (−516 to 510) *p* = 0.992	33.4 (−479 to 546) *p* = 0.898
**Work Satisfaction (Single item 0–10 scale)**	LBP 6.1 (5.1 to 7.1) noLBP 7.2 (6.4 to 8.1)	−1.1 (−2.5 to 0.2) *p* = 0.094	−1.2 (−2.5 to 0.1) *p* = 0.076
** *Pressure Pain Thresholds (kPa)* **			
**Wrist Pre-lifting**	LBP 469.4 (412.5 to 526.3) noLBP 494.6 (414.1 to 575.2)	−25.2 (−125.5 to 75.0) *p* = 0.622	−33.6 (−132.0 to 64.8) *p* = 0.503
**Wrist Post-lifting**	LBP 523.4 (446.5 to 600.3) noLBP 512.1 (435.7 to 588.6)	11.3 (−92.0 to 114.5) *p* = 0.831	3.1 (−103.1 to 109.3) *p* = 0.954
**Change between pre- and post-lift (wrist)**	LBP 54.0 (4.5 to 103.5) noLBP 17.5 (−28.5 to 63.5)	36.5 (−23.9 to 96.9) *p* = 0.236	36.7 (−27.3 to 100.7) *p* = 0.261
**Lower back pre-lifting**	LBP 517.8 (435.2 to 600.4) noLBP 564.2 (454.0 to 674.4)	−46.4 (−177.5 to 84.7) *p* = 0.488	−58.0 (−189.6 to 73.6) *p* = 0.388
**Lower back post-lifting**	LBP 630.3 (528.2 to 732.4) noLBP 623.9 (514.1 to 733.7)	6.4 (−135.5 to 148.3) *p* = 0.930	−14.8 (−157.7 to 128.1) *p* = 0.839
**Change between pre- and post-lift (back)**	LBP 112.5 (60.0 to 164.9) noLBP 59.7 (21.3 to 98.1)	52.8 (−12.6 to 118.2) *p* = 0.114	43.2 (−17.0 to 103.4) *p* = 0.159

* Adjusted for age and sex.

**Table 4 ijerph-20-01903-t004:** Associations between pain ramp during lifting and each non-biomechanical variable in workers with a history of low back pain (LBP group).

Pain RAMP in the LBP Group as Dependent Variable	Unadjusted Coefficient (95% CI) Degrees of Pain Slope ^#^	Adjusted * Coefficient (95% CI) Degrees of Pain Slope
BACK-PAQ	0.003 (−0.005 to 0.010) *p* = 0.492	0.001 (−0.009 to 0.011) *p* = 0.846
DASS—Depression	0.004 (−0.014 to 0.022) *p* = 0.657	−0.004 (−0.024 to 0.016) *p* = 0.704
**DASS—Anxiety**	−0.008 (−0.025 to 0.009) *p* = 0.357	**−0.018 (−0.035 to −0.001)** ***p* = 0.039**
DASS—Stress	0.000 (−0.019 to 0.019) *p* = 1.000	0.000 (−0.019 to 0.019) *p* = 0.976
DASS—Combined score	−0.001 (−0.007 to 0.005) *p* = 0.783	−0.004 (−0.010 to 0.003) *p* = 0.285
Self-efficacy	−0.005 (−0.027 to 0.017) *p* = 0.640	0.008 (−0.027 to 0.043) *p* = 0.652
How safe is bent back lifting?	0.009 (−0.035 to 0.053) *p* = 0.683	0.010 (−0.036 to 0.056) *p* = 0.666
Subjective worry pre-lift task	0.023 (−0.057 to 0.104) *p* = 0.570	0.004 (−0.069 to 0.078) *p* = 0.906
Work satisfaction	0.006 (−0.025 to 0.037) *p* = 0.701	0.007 (−0.031 to 0.046) *p* = 0.707
Health complaints—Modified MSK subscale	0.022 (−0.009 to 0.054) *p* = 0.165	0.008 (−0.038 to 0.054) *p* = 0.733
Health complaints—Pseudoneurology subscale	−0.013 (−0.016 to 0.041) *p* = 0.385	−0.012 (0.060 to 0.036) *p* = 0.627
**Health complaints—GI subscale**	−0.008 (−0.032 to 0.015) *p* = 0.492	**−0.051 (−0.066 to −0.036)** ***p* < 0.001**
Health complaints—Allergy subscale	−0.084 (−0.340 to 0.172) *p* = 0.519	−0.110 (−0.303 to 0.083) *p* = 0.263
Health complaints—Flu subscale	−0.048 (−0.113 to 0.018) *p* = 0.155	−0.036 (−0.127 to 0.055) *p* = 0.441
Sleep	0.004 (0.048 to 0.055) *p* = 0.889	−0.012 (−0.062 to 0.037) *p* = 0.623
IPAQ—Total activity	**<0.001 (0.000 to 0.001)** ***p* = 0.026**	0.000 (−0.000 to 0.000) *p* = 0.462
IPAQ—Sedentary time	0.000 (−0.000 to 0.000) *p* = 0.868	0.000 (−0.000 to 0.000) *p* = 0.998
PPT change at wrist	0.000 (−0.001 to 0.001) *p* = 0.688	0.000 (−0.001 to 0.001) *p* = 0.307
PPT change at back	0.000 (0.000 to 0.001) *p* = 0.283	0.000 (0.000 to 0.001) *p* = 0.293
RMDQ	0.013 (−0.008 to 0.033) *p* = 0.233	−0.003 (−0.043 to 0.036) *p* = 0.863
PCS	0.000 (−0.012 to 0.012) *p* = 0.998	−0.005 (−0.016 to 0.005) *p* = 0.298
Tampa	−0.003 (−0.013 to 0.007) *p* = 0.596	−0.009 (−0.022 to 0.003) *p* = 0.143

* Adjusted for age and sex. ^#^ These coefficients are the change in pain ramp slope (degrees) for each one-point change in the questionnaire or PPT score.

**Table 5 ijerph-20-01903-t005:** The relationship between pain during lifting and the feeling of fatigue during lifting.

	Unadjusted (95%CI)	Adjusted
Fatigue during lifting	0.025 (−0.120 to 0.169) *p* = 0.739	0.020 (−0.124 to 0.165) *p* = 0.781
**LBP/noLBP group (noLBP is the reference category)**	**1.164 (0.512 to 1.816)** ***p* < 0.001**	**1.298 (0.630 to 1.964)** ***p* < 0.001**
**The interaction between fatigue during lifting and the LBP group**	**0.240 (0.057 to 0.423)** ***p* = 0.010**	**0.240 (0.058 to 0.422)** ***p* = 0.010**
**Lift decile (the effect of time)**	**0.009 (0.004 to 0.015)** ***p* = 0.001**	**0.010 (0.004 to 0.015)** ***p* = 0.001**
Age		−0.020 (−0.044 to 0.005) *p* = 0.114
**Sex**		**−0.681 (−1.30 to −0.061)** ***p* = 0.031**
Constant	−0.204 (−0.446 to 0.038)	0.630 (−0.176 to 1.437)

## Data Availability

The data pertaining to the results of this study are available by contacting the corresponding author.

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
