# Peer review of "An Exploration of the Influence of Non-Biomechanical Factors on Lifting-Related LBP"

_ijerph, 2023, doi:10.3390/ijerph20031903_

Round 1

Reviewer 1 Report

Dear Authors,

You can see my comments in the boxes in the main text.

Best regards

Reviewer 2 Report

I would like to congratulate the authors for preparing a very solid article for the 

Comment 1

Please describe in more detail the musculoskeletal examination that took place in the screening process. While it is clear the focus was on non-biomechanical factors, was any data collected on Lumbar or Hip AROM? Were patients screened via myotome, reflex, sensory or neural tension testing to rule out radiculopathies? Were the presence of Trigger points screened during the physical examination of the patients? Often trigger points can display hypersensitivity to pressure testing. Were any of the areas of PPT threshold testing done over the areas of trigger points? If this was not done I recommend placing this information carefully into the limitations section. If the data is available adding a table would be worthwhile.

Reviewer 3 Report

Thank you for permitting me to review this manuscript

abstract 

the secondary objective should be rephrased in the absract and in the introduction  

please develop^multimensional nature of LBP  which is mentionned in the conclusion 

methods 

please elaborate  precisely missing  data 

results 

I do not understand why an outlier BMI should be excluded, was it in the inclusion criteria

younger parients may have  low back pain later as the mean age is 32 and 37 ,  precaution should be taken because of bias 

table 1 : Please recheck age comparison as I found a statistical difference with my software 

table 3 should e bettr formated  in addition , page 11 there is a non alignement 

most of the non biomechanical issues are questionnaire and subjective evaluation which should be signaled in the discussion 

although sedentary life and total activitty was assessed sporting history ws not assessed please discuss

implication do not consider sleep disturbances 

Round 2

Reviewer 1 Report

Dear Authors,

Thank you for the changes, I think the article has become better and is suitable for publication.

I wish conveniences.